# The Cell Wall Characterization of Brown Alga *Cladosiphon okamuranus* during Growth

**DOI:** 10.3390/plants12183274

**Published:** 2023-09-15

**Authors:** Yuka Miwa, Mahanama Geegana Gamage Awanthi, Kouichi Soga, Atsuko Tanaka, Michihiro Ito, Yuichiro Numata, Yoichi Sato, Teruko Konishi

**Affiliations:** 1Department of Bioscience and Biotechnology, Faculty of Agriculture, University of the Ryukyus, Senbaru, Nishihara-cho 903-0213, Okinawa, Japan; 2United Graduate School of Agricultural Sciences, Kagoshima University, Korimoto, Kagoshima-shi 890-0065, Kagoshima, Japan; 3Department of Biology, Graduate School of Science, Osaka Metropolitan University, Sugimoto, Sumiyoshi-ku 558-8585, Osaka, Japan; 4Department of Chemistry, Biology and Marine Science, Faculty of Science, University of the Ryukyus, Senbaru, Nishihara-cho 903-0213, Okinawa, Japan; 5Center of Molecular Biosciences, Tropical Biosphere Research Center, University of the Ryukyus, Senbaru, Nishihara-cho 903-0213, Okinawa, Japan; 6Bio-Resources Business Development Division, Riken Food Co., Ltd., Miyauchi, Tagajo-shi 985-0844, Miyagi, Japan

**Keywords:** Okinawa Mozuku, cell wall polysaccharide, fucoidan

## Abstract

The present study provides new insights into the growth of the brown algal cell wall by showing that cell wall polysaccharides play an important role in the process of growth, considering the physicochemical characteristic of young and old *Cladosiphon okamuranus*. To determine its structural variation in detail, the cell wall was sequentially fractionated into five fractions: hot water (HW), ammonium oxalate, hemicellulose-I (HC–I), HC-II, and cellulose, and analyzed physicochemically. Results showed that almost 80% of the total recovery cell wall from both young and old thalli was HW, and HC-I contained mainly fucoidan composed of Fucose, Glucuronic acid, and sulfate in molar ratios of 1.0:0.3:0.6~0.7 and 1.0:0.3:0.2~0.3, respectively. Fucoidan in HW was a highly sulfated matrix polysaccharide abundance in young thalli, while fucoidan in HC-I was rich in old thalli and functions as hemicellulose in land plants, crosslinking with cellulose and strengthening the cell wall. We found that HW and HC-I were particularly involved in the growth and strength of old thalli appeared to be due to the deposition of HC-I and the reduction in water content during the growth process.

## 1. Introduction

The cell wall is a protective structure surrounding the cell membrane of some types of eukaryotic and prokaryotic cells, including land plants and algae. In land plants, the structure, synthesis, and function of distinct types of cell and species have been identified extensively for decades [1]. The fundamental roles of the plant cell wall involve morphogenesis and growth, providing mechanical strength, defending against biotic and abiotic stresses, adaptation, and so on [2]. The plant cell wall mainly contains different types of polysaccharides, such as cellulose, hemicellulose, and pectin, and a small number of structural proteins. Many experimental results indicate that the cell wall is formed by the successive deposition of pectic substances, hemicellulose, cellulose, and lignin during cell wall differentiation, with the cell wall assembly process itself being irreversible, to form a dynamic network to bear the high intracellular turgor pressure. The polysaccharide in the plant cell wall was found to essentially involve in many facets of growth and development, such as cell expansion polarity and the thickening of cell walls through the deposition of polysaccharides [2].

Like land plants, macroalgae such as red, green, and brown algae consist of a polysaccharide-rich cell wall, which is approximately 4–76% of algal dry weight [3]. Among them, brown algae (Phaeophyta) represent the largest biomass-producing organism, containing more than 250 genera and 1500–2000 species, in the marine ecosystem [4], and have different types of polysaccharides in common with plants (cellulose), animals (sulfated fucans), and some bacteria (alginates) [1,5]. The main polysaccharides in brown algal cell walls are fucoidan and alginates, which encompass up to 45% of algal dry weight, while cellulose only accounts for 1–8% of algal dry weight [6]. Regarding the functionality of the main cell wall polysaccharide of brown algae, alginate fine structure likely contributes to cell wall rigidity, while sulfated polysaccharides, such as fucoidan, probably have a main role in the osmotic regulation [6]. Furthermore, preceding studies of brown algae found that the cell wall contributed to cell adhesion [7], cell expansion [8], cell development with cell differentiation in growing filaments [9], polar axis fixation, and cell fate [10]. Although the cell wall was found to be involved in growth and development, little attention has been paid to date to the involvement of cell wall polysaccharides in growth.

Herburger et al. investigated the cell wall polysaccharide content in charophyte green algae changes during growth [11]. For instance, the pectin content of *Zygnema* and hemicelluloses content in *Klebsormidium* increase with increasing cell age. Furthermore, laminarans and fucoidans content of brown algal cell wall was found to be significantly higher in mature *Laminaria cichorioides* in autumn compared to unripe algae, suggesting that algal polysaccharides would play a key role in cell growth [12]. *Cladosiphon okamuranus*, Okinawa Mozuku in Japanese, is one of the most important edible brown algae in Japan and an excellent source of fucoidan for all brown algae varieties. We found the presence of fucoidan as hemicellulose in the cell wall from *C. okamuranus*, suggesting that this fucoidan might be involved in reinforcing cell wall structure by cross-linking to cellulose [13]. Thus, we hypothesized that cell wall polysaccharides, especially the fucoidan as hemicellulose, had a significant effect on the growth. *Cladosiphon okamuranus* is widely cultivated in Okinawa, Japan, from November to June, and its harvest season generally starts in January and ends in June, although this varies depending on the location. Here, we analyzed the cell wall structure of immature thalli of *C. okamuranus* (young thalli), which was harvested at the beginning of the harvest season, and mature thalli (old thalli), which was harvested at peak season or later. Our present study focuses on how cell wall polysaccharides contribute to growth in brown algae, with special reference to *C. okamuranus*, which is identified as an abundant source of fucoidan for all brown algae. 

## 2. Results

### 2.1. Physical Characteristics of Young and Old Thalli

A variation in physical characteristics was observed between young and old algae samples after harvesting. The young thalli feel stickier or slimier to the touch compared to the old thalli, and the old thalli are darker in color and feel harder than young thalli. To understand the differences in strength between young and old thalli quantitatively, the breaking strength was measured using a tensile tester in both main axis and lateral branches in each sample. The results showed that the breaking strength of the main axis and lateral branches of old thalli was 40 g and 36 g, respectively, which was significantly higher than that of young thalli, as presented in Figure 1b. The insides of the main axes were hollow in both old and young thalli (Figure 2). The main axes were composed of two elements, medullary cell layers and assimilatory filaments. In both thalli, a single cell layer was observed below the basal cell of assimilatory filaments (Figure 2c,d), and the structures of the medullary layer were similar to each other. Regarding the assimilatory filaments, the number of branches from a single basal cell in the young thalli seemed to be more than in the old one, although the actual number was not counted in this study. The different physical properties of young and old thalli could reflect differences in the cell wall components, as these produce diverse physiochemical properties during growth. Hence, the cell walls of each sample were prepared as alcohol-insoluble residue (AIR) and fractionated.

### 2.2. Yield of AIR

The yields of the AIRs prepared from young and old thalli were 2.8% and 5.5%, based on wet weight, respectively, in 2018. The moisture content was 93.2% and 91.4% for young and old thalli, respectively, in 2018. A similar trend was recorded for the yields of AIRs and moisture content of young and old thalli in 2019, as shown in Table 1. These results suggested that once the algal body grows from young to old, the cell wall material content increases while the moisture content decreases. Since we found many different characteristic variations, such as color, strength, moisture, and content of AIR between young and old thalli, our next steps were focused on a detailed structural analysis of the cell walls to understand how these differences occur during the growth of thalli. Therefore, the AIR was fractionated in to five fractions and their chemical composition was evaluated as the next steps to find whether the maturation of the cell wall is involved in the chemical composition variation.

### 2.3. Yield and Composition of Different Cell Wall Fractions

Using sequential chemical extraction, AIR prepared from young and old thalli were fractionated into five fractions: hot water (HW), ammonium oxalate (AO), hemicellulose (HC) -I, HC-II, and cellulose (CL), and their yield displayed in Table 2. Results showed that HW and HC-I were the main fractions, comprising almost 80% of the total cell wall recovered from AIR and AO, and HC-II and CL comprised of the remaining 20% in every sample. However, the ratio between the two main components, HW and HC-I, varied between young and old algal body. For instance, the yield of HW and HC-I in young thalli was 62.4% and 18.1%, respectively, while they were 49.8% and 30.8%, respectively, in old thalli in 2018. The same tendency occurred in 2019, as presented in Table 2. These results imply that yield of HC-I increased while yield of HW decreased, but the total yield of HW and HC-I was almost constant during growth period. Significant variation was not shown in yield of the other minor fractions, AO, HC-II, and CL, between old and young thalli, as shown in Figure 3.

The chemical composition of each cell wall fraction in young and old thalli harvested in 2018 is presented in Table 2. Approximately more than 50% of the dry weight of each fraction was sugar, with the highest Uronic acid (UA) content recorded in the AO fraction in young thalli. It is clear that there was a higher sulfate content (24.5%) in HW compared to all other fractions, and the sulfate content of HC-I was 6.4% in young thalli. Protein was co-extracted together with a minor amount of polyphenol in every fraction, and it was highlighted that the highest content of protein and polyphenol exists in the HC-I fraction in young thalli. The composition analysis of all fractions obtained from old thalli appears to be consistent with the young thalli. In addition, the chemical composition of each cell wall fraction in young and old thalli harvested in 2019 (Table 3) showed the same tendency as the 2018 sample. 

### 2.4. Sugar Composition Analysis

Since there were no more differences in the composition of the young and old thalli, sugar composition analysis was carried out to understand the structural differences in detail. The sugar composition of each cell wall fraction in young and old thalli harvested in 2018 is presented in Table 4. Our previous study found that HW and HC-I mainly contained fucoidan with different structures, while AO was mostly composed of alginate, and CL was almost cellulose [13]. Similarly, the results show that HW and HC-I contained mainly fucoidan, and the molar ratios of Fuc:GlcA:SO_3_^−^ of HW and HC-I were 1.0:0.3:0.6~0.7 and 1.0:0.3:0.2~0.3, respectively, with a trace amount of Gal, Glc, and Xyl in both young and old thalli, as shown in Table 4. The xylose content of fucoidan in HC-I is higher than that of fucoidan in HW in both young and old thalli. The presence of fucoidan was confirmed in all fractions except CL for both young and old thalli. However, the molar ratio of Glc:Fuc was 55.9 in young thalli and 26.7 in old thalli, meaning that the amount of Fuc in old thalli was about twice as high as that in young thalli. Since Fuc is a constituent sugar of fucoidan, it is supposed that fucoidan in is more strongly bound to cellulose in the old thalli cell wall compared to young.

Approximately similar results were reported for the sugar composition of each cell wall fraction in young and old thalli harvested in 2019, as shown in Table 5. Since most of the compounds in the cell wall from *C. okamuranus* were extracted in HW and HC-I and there was no more variation in the chemical composition in the cell wall harvested in two consecutive years, further analysis was focused only on HW and HC-I in young and old thalli harvested in 2019.

### 2.5. Molecular Weight Distribution

Molecular weight distribution of HW and HC-I in young and old thalli was determined using high-performance size-exclusion chromatography, as shown in Figure 4. Each chromatogram mainly comprised one major peak (Mp), which eluted later and more broadly in HC-I than HW. The molecular weight of HW (2.1 × 10^6^) and HC-I (0.7 × 10^6^) in old thalli was higher than that of HW (2.1 × 10^6^) and HC-I (0.5 × 10^6^) in young thalli. These results suggested that the molecular weight of HW and HC-I increased with growing period. The peak at V_t_ was NaCl in the sample and eluting buffer, while the small peak occurred just before V_t_ was comparatively lower (<10%) than Mp in the chromatograms. 

### 2.6. Anion Exchange Chromatography

To identify further structural variation in cell wall during growth, HW and HC-I in young and old thalli were fractionated using anion exchange chromatography. The anion exchange chromatogram of HW from both young and old thalli showed two main peaks, including peak at flow-through (HW-FT) and peak eluted with NaCl (HW-NaCl), as shown in Figure 5a, but variation in the yield of each fraction between young and old thalli was reported. The yield of HW-FT in old thalli (28.2%) was higher than that in young thalli (14.4%), while the yield of HW-NaCl in young thalli (76.4%) was higher than that in old thalli (67.2%). We expected HW-FT to have a lower number of carboxyl and sulfated groups than HW-NaCl, as it eluted as unbound polysaccharide in flow through fraction, but sugar composition analysis revealed that HW-FT and HW-NaCl were mainly composed of constitutes of fucoidan such as Fuc, GlcA, and sulfate, in quite a similar molar ratio. However, the Glc content was higher in HW-FT than HW-NaCl in both young and old thalli (Table 6).

The anion exchange chromatography of HC-I in young and old thalli showed only one peak eluted with NaCl (HC-I-NaCl), although both elution profiles seem to be slightly similar (Figure 5b). The sugar composition analysis of HC-I-NaCl in both young and old thalli revealed that fucoidan was the predominant sugar component, composed of Fuc, GlcA, and sulfate in the molar ratio of 1:0.2~0.4:0.5~0.6, as shown in Table 6. These results suggest that purified HW and HC-I, obtained using anion exchange chromatography, were basically fucoidan, but the yield of HW-FT and HW-NaCl fractions varies between young and old.

## 3. Discussion

Many studies have proven that the polysaccharides in plant cell walls are involved in many essential facets of growth and development, such as cell expansion polarity and the thickening of cell walls through the deposition of polysaccharides [2]. However, little is known about the involvement of polysaccharides in the growth and development of algae. The present study provides new insights into the growth of brown algal cell walls by showing that the cell wall polysaccharides play a leading role in the process of growth, considering the physicochemical characteristics of young and old *C. okamuranus*. Firstly, we examined the physical characteristics and identified the several variations between young and old thalli. The young thalli were lighter in color and felt slimier to the touch compared to the old thalli. It was found that the molecular weight and sulfate content of the fucoidan might be the reason for the variation in the sliminess of thalli harvested at different time [14]. Furthermore, the breaking strength of the main axis and lateral branches of old thalli were significantly higher than that of young thalli (Figure 1b). In anatomical analysis, it was confirmed that the basic structures of the young and old thalli were almost the same except for the assimilatory filaments, suggesting that the generation of tension strength may be influenced by microscale factors such as the thickness and/or composition of the cell wall (Figure 2). Preceding studies have consistently found that old algal tissue was significantly stronger than newly formed tissue, which has been suggested to be due to the differences in tissue composition or structure [15,16,17]. These biomechanical characteristics of the different tissues would corelate with prominent differences in moisture content, cortical layer thickness, cell wall thickness, and the number of insoluble polysaccharides such as cellulose, hemicellulose, and fucoidan [15]. According to the results of our study, the moisture content was lower in old thalli (91.4%) than young (93.2%), while yield of AIR was higher in old thalli (5.5%) than young thalli (2.8%) (Table 1), suggesting that cell wall material content increases and moisture content decreases during growth. A similar trend was reported in moisture content and crude fucoidan content after March in *C. okamuranus* and kelp species such as *L. digitata*, *L. hyperborea*, *Saccharina latissima*, and *Alaria esculenta* [14,18]. Therefore, it can be proven that algal tissue becomes stronger in line with the deposition of cell wall materials and the reduction in water content during the growth process. 

Since increase in the cell wall material was observed with the growth of thalli, the variation in the cell wall composition needs to be analyzed to understand which polysaccharides mostly engage with the process of growth. Therefore, AIR was fractionated into five fractions: HW, AO, HC-I, HC-II, and CL. The results show that almost 80% of the total recovery cell wall from AIR was HW and HC-I, and the amount of HC-I increased, while HW decreased during growth (Table 2 and Table 3). Our previous study found that HW and HC-I mainly contained fucoidan composed of Fuc, GlcA, and sulfate in molar ratios of 1.0:0.3:0.9 and 1.0:0.2:0.3, respectively, and their structure differed in terms of contents of sulfate and Xyl, MW, and profile of the small angle x-ray scattering [13]. Highly sulfated polysaccharides in HW were typically fucoidan, and have been suggested to be weakly held in the cell wall matrix and possibly involved in osmotic regulation in brown algae [19], while HC-I contained 1,4-linked Xyl and 1,4-linked Fuc, as well as components of typical fucoidan, suggesting that these are likely to be 1,4-xylan and/or 1,4-fucan, which may be involved in reinforcing cell wall structure by cross-linking to cellulose as hemicellulose in terrestrial plants, keeping CL microfibrils separated, and controlling cell wall expansion [20]. This hypothesis is further supported by the results of this study. The young thalli are rich in matrix polysaccharides; HW and the old thalli possessed a larger amount of fibrillar polysaccharides, HC-I, compared to young thalli, suggesting that cell wall thickening in *Cladosiphon okamuranus* during growth is characterized by an increase in the HC-I content. A similar scenario was reported in land plants, but with different types of polysaccharide. For instance, pectic substances are mostly deposited at a young stage, while the contents of compositional sugars from cellulose and hemicelluloses increase with the maturation of the secondary wall [21]. However, substantial variation was not found in the yield of other minor fractions, AO, HC-II, and CL, between old and young. In contrast, variations in the cellulose amount have been shown to cause changes in both the strength and rigidity of plant tissues [22,23]. Cellulose is the most abundant major polysaccharide, representing up to 50% of terrestrial plants’ cell walls, but only makes up 5% of the cell wall in brown algae, both young and old thalli, in this study [6].

The further structural analysis of HW and HC-I was conducted related to young and old thalli. Sugar composition analysis showed that HW and HC-I in young and old thalli contained mainly fucoidan composed of Fuc, GlcA, and sulfate in molar ratios of 1.0:0.3:0.6~0.7 and 1.0:0.3:0.2~0.3, respectively, with a trace amount of Gal, Glc, and Xyl. Although there were variations in the sulfate and xylose content between HW and HC-I, there were no significant differences observed in the sugar composition and sulfate content between young and old thalli. However, fucoidan appears to be more strongly bound to cellulose in the old thalli cell wall compared to the young thalli cell wall according to the Glc:Fuc molar ratio in the CL fraction (Table 4 and Table 5). We believe that a large quantity of the fucoidan in HC-I in old thalli might be bound strongly to the cellulose. Moreover, HW and HC-I in young and old thalli were further characterized by the determination of their molecular weight, which was found to be larger in old thalli than young. Zvyagintseva et al. also found that the molecular weight of fucoidan in 2-year-old *L. cichorioides* was significantly higher (20–30 kDa) than a 0.8-year-old sample (8–10 kDa) [12]. It appears that the high molecular weight of old thalli is to be due to an increase in its degree of polymerization during growth [24]. Further fractionation was carried out for HW and HC-I in both young and old thalli using anion exchange chromatography to explore their purity and heterogeneity based on their charge density. These results further indicated that purified HW contains two main fractions, HW-FT and HW-NaCl, but that the yield of HW-NaCl fractions in young thalli were higher than that of old, while HC-I eluted only HC-I-NaCl, and both young and old thalli appeared to be similar (Table 6). These results confirmed that there were many structural variations in the main fractions, HW and HC-I, which may be related to the localization of young and old thalli, which is involved in the process of growth.

## 4. Materials and Methods

### 4.1. Algal Sample

*Cladosiphon okamuranus,* cultivated in Chinen, Nanjo City, Okinawa Prefecture (26°08′05′′ N, 127°48′19′′ E), was harvested after about 40 days (young thalli) and about 90 days (old thalli), after planting in the sea, in 2018 and 2019, respectively.

### 4.2. Measurement of Moisture Content

About 0.1 g of sample was taken after harvesting, and moisture content was measured using a moisture content analyzer (MOC-63U SHIMADZU, Kyoto, Japan).

### 4.3. Measurement of Tensile Strength

The main axis was defined as the thickest branch growing from the base of the thalli, and the lateral branches were defined as branches departing from the main axis, as shown in Figure 1a. The tensile strength of the main axis and lateral branches was determined using a tensile tester (STB1225S, A&D Co., Ltd., Tokyo, Japan). About 10 mm of thalli was fixed between two clamps and stretched by raising the upper clamp at a speed of 20 mm/min until the thalli broke.

### 4.4. Light Microscopy

The samples fixed with 100% ethanol were used for making cross sections without washing. The samples were observed through a light microscope (BX53F2, OLYMPUS, Tokyo, Japan) equipped with a digital camera (WRAYCUM NOA2000, WRAYMER, Osaka, Japan).

### 4.5. Preparation of Alcohol Insoluble Residue (AIR)

Each algal sample (50 g) was ground using a blender in 4 volumes of ethanol (200 mL), followed by centrifugation at 8000× *g* at 25 °C for 15 min; then, the precipitate was sequentially treated with 80% ethanol, 100% ethanol, and methanol:chloroform (1:1, v:v), followed by acetone. Suction filtration was performed and residue was dried at room temperature and used as AIR [13].

### 4.6. Fractionation of Cell Wall Polysaccharides

The fractionation procedure was based on the different solubilities of the polysaccharides from brown seaweeds, as described in our previous study [14]. Briefly, AIR was sequentially treated with hot water (HW), 0.25% ammonium oxalate (AO), 4% KOH, and 24% KOH to produce 4 fractions: HW, AO, hemicellulose-I (HC-I), and HC-II. The final residue was collected as cellulose (CL). After neutralizing HC-I and HC-II using acetic acid, AO, HC-I, and HC-II were dialyzed, lyophilized, and used for analysis. The CL was washed with water after neutralizing with acetic acid, and then lyophilized.

### 4.7. Chemical Composition Analysis

Total sugar and UA were determined using the phenol–sulfuric acid method using Fuc as a standard [25], and the *m*-hydroxybiphenyl method using GlcA as a standard [26], respectively. Total polyphenols were quantified using the Folin–Ciocalteu method, using gallic acid as a standard [27]. Protein content was measured using bicinchoninic acid assay (BCA) following the manufacturer’s instructions of a BCA Protein Assay Kit (Takara Bio Inc., Shiga, Japan). The calibration curve was prepared using bovine serum albumin.

To estimate SO_3_^−^, the sample was hydrolyzed in 2 M trifluoroacetic acid at 121 °C for 1 h, hydrolysate was subjected to high-performance liquid chromatography with an AS4A-SC column (4 mm × 250 mm, Dionex Co., Tokyo, Japan). The column was eluted at 1.5 mL/min at room temperature with a buffer containing 1.7 mM NaHCO_3_ and 1.8 mM Na_2_CO_3_ [28]. The SO_3_^−^ content in the sample was calculated from a calibration curve using Na_2_SO_4_ as a standard [13].

### 4.8. Sugar Composition Analysis

To analyze sugar composition, samples were hydrolyzed to be used for SO_3_^−^ estimation, while the AIR and CL were treated with ice-cold 72% (*w*/*w*) H_2_SO_4_ at 4 °C for 1 h with sonication, followed by hydrolysis with 2 N H_2_SO_4_ at 121 °C for 1 h [29,30]. Monosaccharide in the hydrolysate was analyzed using high-performance anion exchange chromatography coupled with a pulsed amperometric detector (HPAEC-PAD), using a Carbo Pac PA1 column (4 mm × 250 mm, Thermo Fisher Scientific, MA, USA). The column was eluted at a flow rate of 1 mL/min at 35 °C with 14 mM NaOH for neutral sugar, followed by a linear gradient program of 0–250 mM CH_3_COONa in 100 mM NaOH for UA.

### 4.9. Determination of Molecular Weight (MW)

The MW of HW and HC-I were determined by size-exclusion chromatography (LC-6A; Shimadzu Co., Kyoto, Japan) equipped with a TSKgel G5000 PWXL column (7.8 mm × 300 mm, Tosoh Co., Tokyo, Japan) and a refractive index detector RID-10A [31]. The column was eluted by 0.2 M NaCl at a flow rate of 0.3 mL/min at 40 °C. Pullulan P-10 (MW = 0.96 × 10^4^), P-50 (4.71 × 10^4^), P-200 (20.0 × 10^4^), and P-800 (70.8 × 10^4^) (Showa Denko Co., Tokyo, Japan) were used as standards.

### 4.10. Anion Exchange Chromatography

For further analysis of HW and HC-I extracted from young and old thalli, anion exchange chromatography was used. A total of 50 mg of each fraction was dissolved in 20 mM Tris-HCl (pH 7.4) and applied onto a DEAE–Sephacel column (2.5 × 10 cm, GE Healthcare, Uppsala, Sweden) equilibrated with the same buffer. The column was washed with 20 mM Tris-HCl (pH 7.4) and then eluted with a linear gradient of 0–3.5 M NaCl in 20 mM Tris-HCl (pH 7.4), followed with 3.5 M NaCl in the same buffer. 

### 4.11. Statistical Analysis

Data were expressed as mean ± standard deviation of three determinations. Statistical comparison was performed via a one-way analysis of variance followed by Tukey’s test. Probability values of less than 0.05 (*p* < 0.05) were considered significant.

## 5. Conclusions

In conclusion, differences in the physical and chemical composition of young and old thalli of *C. okamuranus* were observed related to their growth. Regarding the characteristics of thalli, the old thalli were stronger, darker in color, and felt less slimy to the touch compared to the young thalli. The moisture content was lower in old thalli than young thalli, while the yield of AIR was higher in old thalli than young thalli. Almost 80% of the total recovery cell wall from both young and old thalli was HW, and HC-I contained mainly fucoidan composed of Fuc, GlcA, and sulfate in molar ratios of 1.0:0.3:0.6~0.7 and 1.0:0.3:0.2~0.3, respectively. However, the ratio of HW and HC-I was different between young and old thalli; in old thalli, the yield of HC-I increased while the yield of HW decreased. These results suggested that the cell wall structure of *C. okamuranus* changes during growth, such as the ratios of the two main components, HW and HC-I. Furthermore, it was suggested that the strength of thalli during growth was significantly affected by HC-I.

## Figures and Tables

**Figure 1 plants-12-03274-f001:**
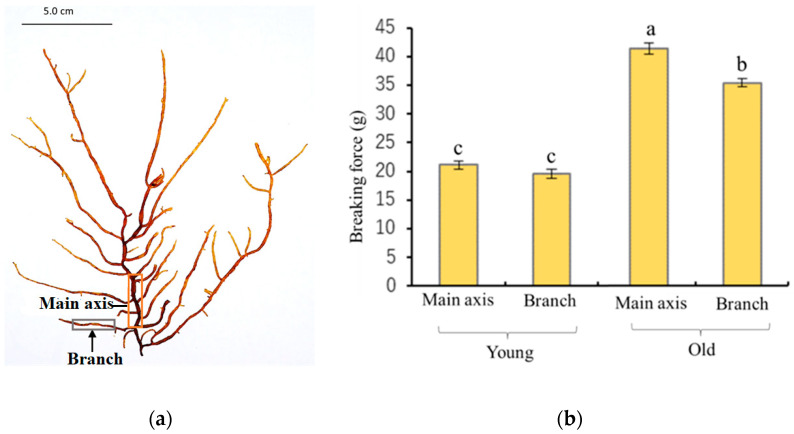
(**a**) Schematic view of *Cladosiphon okamuranus* and its main axis and lateral branches. (**b**) Hardness (breaking strength) of the main axis and lateral branches of young and old *Cladosiphon okamuranus*. Different letters indicate a significant difference (*p* < 0.01; Tukey–Kramer test) between samples (*n* = 20).

**Figure 2 plants-12-03274-f002:**
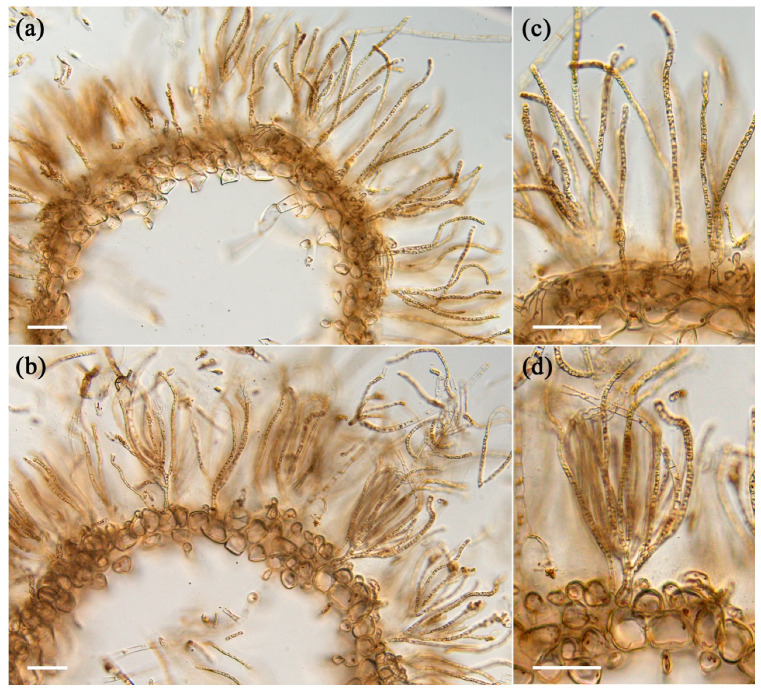
Cross sections of young and old *Cladosiphon okamuranus*. (**a**,**b**) The main axes become hollow in both old (**a**) and young (**b**) thalli. Anatomically, there are no differences in the number of medullary layers and in the cell shape of medullary cells. (**c**,**d**) Magnified images of (**a**) and (**b**), respectively. The numbers of branches of assimilatory filaments from a basal cell seem to vary between old (**c**) and young (**d**) thalli. Scales; 100 μm.

**Figure 3 plants-12-03274-f003:**
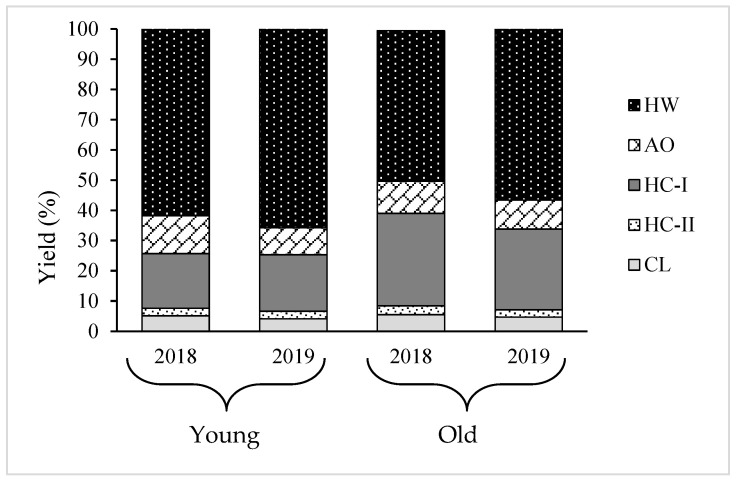
Yield of different polysaccharide fractions extracted from young and old thalli of *C. okamuranus*: HW, Hot water; AO, Ammonium oxalate; HC-I, Hemicellulose-I; HC-II, Hemicellulose-II; CL, Cellulose.

**Figure 4 plants-12-03274-f004:**
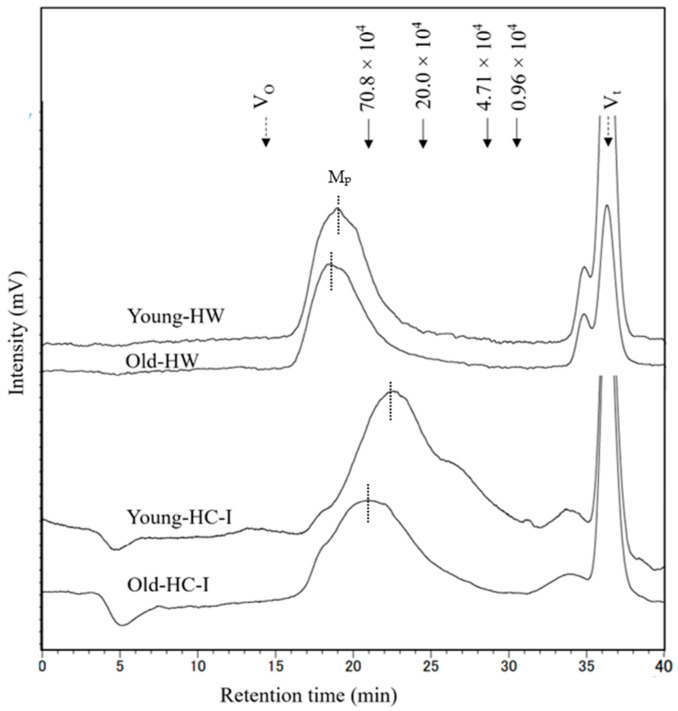
The molecular weight distribution of young-HW, old-HW, young-HC-I and old HC-I harvested in 2019. M_p_ is the main peak at each chromatogram marked by dash line. Solid arrows indicate the elution positions of size standards of pullulan with molecular weight of highest peak. Dashed arrows indicate the void volume (V_o_) and total column volume (V_t_).

**Figure 5 plants-12-03274-f005:**
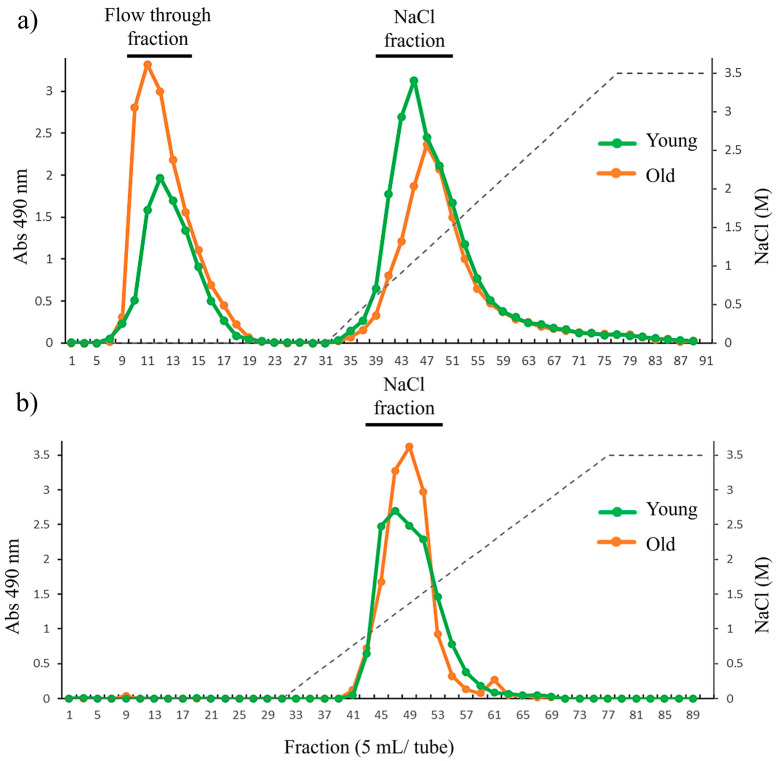
Anion exchange chromatography of (**a**) HW and (**b**) HC-I harvested in 2019. Dotted line across the chromatogram indicates NaCl concentration.

**Table 1 plants-12-03274-t001:** Moisture content and yield of AIR in young and old *C. okamuranus* harvested in 2018 and 2019.

	Harvested Year	Yield of AIR *	Moisture Content *
Young	2018	2.8	93.2
	2019	3.2	92.9
Old	2018	5.5	91.4
	2019	5.0	91.9

* Percentage in wet algae.

**Table 2 plants-12-03274-t002:** Yield and chemical composition of cell wall fractions from *C. okamuranus* harvested in 2018.

	Fraction	Yield ^a^	Total Sugar	UA ^b^	SO_3_^−^	Protein	Polyphenol
	HW	62.4	49.4	24.5	24.5	8.9	2.9
	AO	12.5	59.5	34.3	3.5	6.6	1.7
Young	HC-I	18.1	49.9	22.3	6.4	31.5	8.4
	HC-II	2.5	92.3	7.8	1.9	8.1	5.6
	CL	5.1	100.0	tr	tr	tr	tr
	HW	49.8	56.2	22.9	21.7	14.5	3.9
	AO	10.6	64.8	30.5	6.1	8.0	2.1
Old	HC-I	30.6	62.9	21.1	9.9	22.9	7.3
	HC-II	2.9	80.1	13.7	5.9	10.0	2.4
	CL	5.5	100.0	tr	tr	tr	tr

^a^ Percentage of the total weight fraction recovered in AIR (wt%). ^b^ Percentage weight of Uronic acid in total sugar; other constituents in weight % of the respective fraction. tr, trace amount less than 0.1.

**Table 3 plants-12-03274-t003:** Yield and chemical composition of cell wall fractions from *C. okamuranus* harvested in 2019.

	Fraction	Yield ^a^	Total Sugar	UA ^b^	SO_3_^−^	Protein	Polyphenol
	HW	66.7	55.3	25.6	15.7	15.0	4.3
	AO	8.9	59.2	30.7	3.8	7.4	2.4
Young	HC-I	18.7	67.4	21.8	6.7	21.9	6.8
	HC-II	2.5	114.0	9.8	2.9	5.1	1.2
	CL	4.1	100.0	tr	tr	tr	tr
	HW	58.5	61.6	23.7	20.6	12.1	4.5
	AO	9.6	61.6	31.4	10.4	8.4	2.6
Old	HC-I	26.7	65.3	21.2	8.5	18.9	5.3
	HC-II	2.4	112.1	11.5	11.5	6.8	2.5
	CL	4.7	100.0	tr	tr	tr	tr

^a^ Percentage of the total weight fraction recovered in AIR (wt%). ^b^ Percentage weight of Uronic acid in total sugar; other constituents in weight % of the respective fraction. tr, trace amount less than 0.1.

**Table 4 plants-12-03274-t004:** Sugar compositional analysis of cell wall fractions from *C. okamuranus* harvested in 2018.

	Fraction	Neutral Sugar						UA			SO_3_^−^
		Fuc	Gal	Glc	Man	Xyl		GlcA	GalA/GulA	ManA	
	HW	1.0	tr	tr	-	tr		0.3	tr	-	0.7
	AO	1.0	0.1	tr	tr	0.1		tr	0.3	0.3	0.1
Young	HC-I	1.0	tr	0.1	tr	0.2		0.3	tr	-	0.2
	HC-II	1.0	-	10.8	0.4	0.6		0.9	-	-	0.6
	CL	1.0	-	55.9	-	3.7		-	-	-	-
	HW	1.0	tr	tr	-	tr		0.3	tr	-	0.6
	AO	1.0	tr	-	-	tr		tr	0.3	0.1	0.2
Old	HC-I	1.0	tr	tr	-	0.1		0.3	tr	-	0.3
	HC-II	1.0	-	1.3	0.1	0.3		0.4	-	-	0.4
	CL	1.0	-	26.7	-	0.8		-	-	-	-

tr, trace amount less than 0.1; -, not detected (molar ratio); Fuc, Fucose; Gal, Galactose; Glc, Glucose; Man, Mannose; Xyl, Xylose; GlcA, Glucuronic acid; GalA, Galacturonic acid; GulA, Guluronic acid; and ManA, Mannuronic acid.

**Table 5 plants-12-03274-t005:** Sugar compositional analysis of cell wall fractions from *C. okamuranus* harvested in 2019.

	Fraction	Neutral Sugar						UA			SO_3_^−^
		Fuc	Gal	Glc	Man	Xyl		GlcA	GalA/GulA	ManA	
	HW	1.0	tr	0.1	-	-		0.3	tr	-	0.5
	AO	1.0	tr	tr	0.1	0.1		tr	0.2	0.2	0.1
Young	HC-I	1.0	tr	tr	tr	tr		0.3	tr	-	0.2
	HC-II	1.0	-	12.7	3.1	3.1		2.0	-	-	1.4
	CL	1.0	0.8	71.5	-	-		-	-	-	-
	HW	1.0	tr	0.1	-	tr		0.3	tr	-	0.7
	AO	1.0	tr	tr	-	tr		tr	0.2	0.2	0.4
Old	HC-I	1.0	tr	0.1	-	0.1		0.3	tr	-	0.5
	HC-II	1.0	-	4.3	0.1	0.6		0.7	-	-	1.7
	CL	1.0	-	18.9	-	1.2		-	-	-	-

tr, trace amount less than 0.1; -, not detected (molar ratio); Fuc, Fucose; Gal, Galactose; Glc, Glucose; Man, Mannose; Xyl, Xylose; GlcA, Glucuronic acid; GalA, Galacturonic acid; GulA, Guluronic acid; and ManA, Mannuronic acid.

**Table 6 plants-12-03274-t006:** Sugar compositional analysis of flow through (FT) and NaCl fractions obtained from anion exchange chromatography of young and old HW and HC-I from *C. okamuranus* harvested in 2019.

Fraction		Fuc	Gal	Glc	Man	Xyl	GlcA	SO_3_^−^
	HW-FT	1.0	tr	0.6	-	-	0.3	0.4
Young	HW-NaCl	1.0	tr	tr	tr	tr	0.2	0.4
	HC-I-NaCl	1.0	0.1	0.1	0.1	tr	0.4	0.5
	HW-FT	1.0	-	0.3	-	-	0.3	0.4
Old	HW-NaCl	1.0	tr	tr	tr	tr	0.2	0.3
	HC-I-NaCl	1.0	tr	tr	0.1	tr	0.2	0.6

tr, trace amount less than 0.1; -, not detected (molar ratio).

## Data Availability

All data supporting the conclusions of this article are included in this article.

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
