# Peer review of "The Cell Wall Characterization of Brown Alga Cladosiphon okamuranus during Growth"

_plants, 2023, doi:10.3390/plants12183274_

Round 1

Reviewer 1 Report

In the manuscript "The cell wall characterization of brown alga Cladosiphon okamuranus during growth", the cell wall structure of immature (young) thalli of C. okamuranus which was harvested at beginning of the harvest season and mature (old) thalli which was harvested at peak season or later was analyzed. The study focused on how cell wall polysaccharides contribute to the growth in brown algae with special reference to C. okamuranus which was identified as an abundant source of fucoidan among any brown algae spp. This study is an interesting and clear with a valid and better selection of samples. The study included well-presented data and analysis, and pictures, tables, and figures are clarified. However, minor revisions are needed as follows:

- In general, please check the manuscript for grammatical errors and typos.

- Line 27: please write the full names of Fuc and GlcA at the first mention.

- Line 32: …… the reduction of water content …….

- Keywords: Please do not use the same words in the title to repeat them in the keywords.

- Line 60-61: There is a repetition of the word "cell", please correct.

- Line 65: please move the reference number [11] from line 67 to relate to "Herburger et al" at the beginning of line 65.

- Line 71: Is there a repetition of "Japan"?

- Line 74-79: It is preferable to write the research hypothesis.

- Line 74-79: It is preferable to define what is original in the research and write it directly before the aim and hypothesis of the research in the introduction section.

- Line 314-316: Please write the latitude and longitude of the place of the C. okamuranus cultivation.

- Line 319: Please end the sentence with a full stop.

Moderate revision is required.

Author Response

Comments and Suggestions for Authors

In the manuscript "The cell wall characterization of brown alga Cladosiphon okamuranus during growth", the cell wall structure of immature (young) thalli of C. okamuranus which was harvested at beginning of the harvest season and mature (old) thalli which was harvested at peak season or later was analyzed. The study focused on how cell wall polysaccharides contribute to the growth in brown algae with special reference to C. okamuranus which was identified as an abundant source of fucoidan among any brown algae spp. This study is an interesting and clear with a valid and better selection of samples. The study included well-presented data and analysis, and pictures, tables, and figures are clarified. However, minor revisions are needed as follows:

- In general, please check the manuscript for grammatical errors and typos.

-->  Thank you for your valuable comments. We checked all and changed them one by one. Also, we will ask for English editing again.

- Line 27: please write the full names of Fuc and GlcA at the first mention.

--> We changed them to Fucose and Glucuronic acid, respectively.

- Line 32: …… the reduction of water content …….

--> We corrected “reducing” to “the reduction”.

- Keywords: Please do not use the same words in the title to repeat them in the keywords.

--> We removed Cladosiphon okamuranus and Growth from “Keywords”.

- Line 60-61: There is a repetition of the word "cell", please correct.

--> Sorry. They are individual words, so each “cell” is not able to be deleted.

- Line 65: please move the reference number [11] from line 67 to relate to "Herburger et al" at the beginning of line 65.

--> We put [11] in a suitable place.

- Line 71: Is there a repetition of "Japan"?

--> Sorry. We think all “Japan” is required for explanation, so we cannot delete them.

- Line 74-79: It is preferable to write the research hypothesis.

Line 74-79: It is preferable to define what is original in the research and write it directly before the aim and hypothesis of the research in the introduction section.

--> We added the following sentence in L.75-79.

“We found the presence of fucoidan as hemicellulose in the cell wall from C. okamuranus, suggesting that this fucoidan might be involved in reinforcing cell wall structure by cross-linking to cellulose [13]. Thus, we hypothesized that cell wall polysaccharides, especially the fucoidan as hemicellulose had a significant effect on the growth.”

- Line 314-316: Please write the latitude and longitude of the place of the C. okamuranus cultivation.  

--> We added the latitude of the place as the following, (26°08’05’’N, 127°48’19’’E)

- Line 319: Please end the sentence with a full stop.

--> We added a period at the end of the sentence.

Reviewer 2 Report

Dear Authors,

The manuscript by Miwa et al. “The cell wall characterization of brown alga Cladosiphon okamuranus during growth”, presents the physicochemical characteristic and structural variation of cell walls of young and old Cladosiphon okamuranus. This study is really important because the studied brown algae is not only consumed as food but is also a valuable source of fucoidan used in medicine. Although fucoidans have been known for a long time, not all of their structural features have been elucidated with sufficient certainty, especially their changes during ontogeny, which may affect the yield.I only have two relatively minor criticisms (factually they are only comments/suggestions).

1.      The same data are often repeated in the text, tables and figures.

2.      The Conclusion would have liked more general information with some conclusions about the structure of polysaccharides and their interactions with each other in cell walls of Cladosiphon okamuranus, rather than a summary of the Discussion.

As minor comments, I might add that there are a few inaccuracies in the References list:

Ref. 9 ― ETOILE (capitalize italics (gene name))

Ref. 10 ― journal title?

Ref. 11 ― journal title

Ref. 21 ― journal title

Ref. 27 ― italics of the Latin name

Author Response

Dear Authors,

The manuscript by Miwa et al. “The cell wall characterization of brown alga Cladosiphon okamuranus during growth”, presents the physicochemical characteristic and structural variation of cell walls of young and old Cladosiphon okamuranus. This study is really important because the studied brown algae is not only consumed as food but is also a valuable source of fucoidan used in medicine. Although fucoidans have been known for a long time, not all of their structural features have been elucidated with sufficient certainty, especially their changes during ontogeny, which may affect the yield.I only have two relatively minor criticisms (factually they are only comments/suggestions).

  1. The same data are often repeated in the text, tables and figures. 

-->  Thank you for your comments. We will ask for English editing again.

  1. The Conclusion would have liked more general information with some conclusions about the structure of polysaccharides and their interactions with each other in cell walls of Cladosiphon okamuranus, rather than a summary of the Discussion.

-->  Thank you for your valuable comments. We revised conclusion as the following in L.405-416,

“In conclusion, differences in physical and chemical composition in young and old thalli of C. okamuranus were observed related to their growth. As characteristic of thalli, the old thalli were stronger, darker in color and felt less slimy to the touch compared to the young thalli. Moisture content was lower in old thalli than young, while yield of AIR was higher in old thalli than young. Almost 80% of the total recovery cell wall from both young and old thalli was HW and HC-I contained mainly fucoidan composed of Fuc, GlcA, and sulfate in molar ratios of 1.0: 0.3: 0.6~0.7 and 1.0: 0.3: 0.2~0.3, respectively. However, the ratio of HW and HC-I was different between young and old thalli, which yield of HC-I increased, while the yield of HW decreased in old thalli. These results suggested that C. okamuranus modified cell wall structure such as HW and HC-I ratio during growth. Furthermore, it was suggested that the strength of thalli during growth was significantly affected by HC-I.”

As minor comments, I might add that there are a few inaccuracies in the References list:

-->  We rechecked the list, and corrected errors other than the ones you mentioned.

Ref. 9 ― ETOILE (capitalize italics (gene name))

Ref. 10 ― journal title?

Ref. 11 ― journal title

Ref. 21 ― journal title

Ref. 27 ― italics of the Latin name

Reviewer 3 Report

This paper characterizes the cell wall compositions in a brown algae species during two growth stages. The experiments are properly designed and the data are analyzed and presented nicely. I have no further suggestions.

Author Response

Thank you for your review. 

Reviewer 4 Report

The manuscript by Miwa et al. is in general clearly and well written. The authors analyzed the cell wall components of brown alga at different growth stages and concluded that HC-I enrichment changed the cell wall structure and thus affected the water content of the cells.

I din't find major flaws of the paper but have a few minor comments.

1) in the abstract, the abbreviation Fuc and GlcA was not explanied in the earlier text

2) is there a reason why only 2018 and 2019 data were used and presented? Do you have any data after 2020?

3) there are some grammar issues here and there. For example, Line 128, "Results show that...." should be "Results showed that....". Proofreading will be necessary. 

The language is fine.

Author Response

The manuscript by Miwa et al. is in general clearly and well written. The authors analyzed the cell wall components of brown alga at different growth stages and concluded that HC-I enrichment changed the cell wall structure and thus affected the water content of the cells.

I didn't find major flaws of the paper but have a few minor comments.

1) in the abstract, the abbreviation Fuc and GlcA was not explained in the earlier text

--> Thank you for your comments. We changed them to Fucose and Glucuronic acid, respectively.

2) is there a reason why only 2018 and 2019 data were used and presented? Do you have any data after 2020?

--> We started this research in 2018, and analyzed the immature and mature thalli, respectively. As the results for 2 years, we obtained similar data between 2018 and 2019 as we described in this manuscript. We also analyzed immature and mature thalli in 2020 with moisture content and the yield and chemical composition of cell wall fraction. As the results, samples in 2020 showed the same tendency as 2018 and 2019 samples, showing that the data reproducibility was confirmed. Therefore, we didn’t analyze more details such as sugar composition analysis, ion-exchange chromatography.

3) there are some grammar issues here and there. For example, Line 128, "Results show that...." should be "Results showed that....". Proofreading will be necessary. 

We corrected it. We will ask for English editing again.
